# Evaluation of Malondialdehyde Level, Total Oxidant/Antioxidant Status and Oxidative Stress Index in Colorectal Cancer Patients

**DOI:** 10.3390/metabo12111118

**Published:** 2022-11-15

**Authors:** Karolina Janion, Joanna Katarzyna Strzelczyk, Katarzyna Weronika Walkiewicz, Krzysztof Biernacki, Angelika Copija, Elżbieta Szczepańska, Ewa Nowakowska-Zajdel

**Affiliations:** 1Department of Nutrition-Related Disease Prevention, Department of Metabolic Disease Prevention, Faculty of Health Sciences in Bytom, Medical University of Silesia in Katowice, 41-902 Bytom, Poland; 2Department of Medical and Molecular Biology, Faculty of Medical Sciences in Zabrze, Medical University of Silesia in Katowice, 41-808 Zabrze, Poland; 3Department of Clinical Oncology, No. 4 Provincial Specialist Hospital in Bytom, 41-902 Bytom, Poland; 4Department of Human Nutrition, Faculty of Health Sciences in Bytom, Medical University of Silesia in Katowice, 41-808 Zabrze, Poland

**Keywords:** oxidative stress index, redox balance, lipid peroxidation, colorectal cancer

## Abstract

Oxidative stress may play an important role in colorectal cancer (CRC). The present study included 94 adult patients with CRC (52 men and 42 women) and 26 hospitalized patients (12 men and 14 women) in whom CRC was excluded (control group). During hospitalization, blood serum samples were collected from both groups. Apart from that, anthropometric measurements were taken and other clinical data were analyzed. Serum malondialdehyde (MDA) level, total oxidant status (TOS), total antioxidant status (TAS) and oxidative stress index (OSI) were assayed. Subsequently, the relationship between the analyzed oxidative stress markers and selected clinical characteristics was investigated in both groups. The evaluation of oxidative stress marker values demonstrated that MDA and TAS levels were significantly higher in the control group than the CRC group (*p* < 0.001 and *p* = 0.019, respectively), while TOS levels were significantly higher in the CRC group than the control group (*p* = 0.005). Significantly lower OSI levels were found in the control group than in the CRC group (*p* < 0.001). Similar results can be observed when performing ROC analysis (receiver operating characteristic curve). Preliminary statistical analysis demonstrated that MDA levels in the study group depend on the location of the primary tumour (*p* = 0.035). Based on the post hoc Tukey test, a relationship was demonstrated between the MDA level and the left and right side of the colon (*p* = 0.040). The results may be evidence for a higher level of oxidative stress, including a compromised antioxidative defence system, in patients with CRC.

## 1. Introduction

According to the International Agency for Research on Cancer GLOBOCAN data-base, colorectal cancer (CRC) is one of the most commonly diagnosed cancers in the world. In 2018, more than 1,845,000 new cases were recorded, with an increase in the incidence of the disease in regions with a high social development index. In many countries, despite improved diagnostic methods, CRC mortality has been increasing. Only in 2018, more than 860,000 individuals died of CRC worldwide. The prevalence of CRC increases with age. Over ¾ of cases in both sexes are found in the elderly [1,2].

The alarming statistics concerning the illness have caused a growing interest of its pathogenesis among researchers. As a result, numerous attempts have been made at a more precise exploration of CRC risk factors. It is estimated that as many as 70% of all cases of CRC are sporadic forms of the disease with environmental factors and lifestyle being responsible. Multiple factors, such as poor diet, sedentary lifestyle, obesity, excess alcohol consumption, smoking and inflammatory bowel diseases (ulcerative colitis, Crohn’s disease) as well as hereditary and genetic factors can result in, among others, the overproduction of reactive oxygen species (ROS), leading to compromised redox balance and oxidative stress, which is one of the proposed mechanisms of CRC development [3,4].

The creation of free oxygen radicals is the consequence of both physiological and also pathological processes. Oxygen metabolism, four-electron reduction of molecular oxygen to H_2_O, takes place in the mitochondrial respiratory chain. However, the efficiency of the respiratory chain is estimated to be approximately 96–99%. This means that the remaining approximate 1–4% is subject to only partial reduction. The products of such reactions are called reactive oxygen species (ROS), which include, among others, superoxide radical, hydrogen per-oxide and hydroxyl radical. Under physiological conditions, ROS are quickly neutralized by antioxidant defence mechanisms. However, in pathological situations, the redox balance can be disturbed and oxidative stress can occur [5].

Oxidative stress is defined as an imbalance between the production of ROS and the body’s ability to detoxicate itself and repair the damage caused by ROS, including free radicals. Increased oxidative stress is a threat to the stability of cellular structures. The consequence of ROS excess is damage to macromolecules such as nucleic acids, proteins and lipids, and disturbed metabolic activity of the body, leading to the development of many diseases, including CRC patients [6,7]. 

ROS overproduction is also observed in cancer; however, the mechanisms responsible for inducing oxidative stress in cancer cells have not been fully discovered [4]. ROS can take part in inducing DNA damage, inhibiting apoptosis and promoting the proliferation and metastasis of malignant cells. Prooxidant-antioxidant homoeostasis is maintained primarily through endogenous enzymatic mechanisms; nevertheless, exogenous factors such as lifestyle, diet, medication, dietary supplements and stress are also important for redox balance [6,8].

The majority of researchers have measured only one or a few serum oxidants and/or antioxidants in order to determine the relationship between oxidative stress and CRC. A significant limitation to this type of studies is their selective nature. They do not take into account the presence of other, still unknown serum oxidants or antioxidants and the potential mutual interactions between those compounds, which could have a synergistic effect [9]. The evaluation of the total oxidant status (TOS) and total antioxidant status (TAS) is a more precise approach. These parameters are used to estimate the general oxidation status and antioxidant ability of the body, respectively. Oxidative stress index (OSI) is used for a comprehensive evaluation of redox balance. OSI is defined as a TOS-to-TAS ratio [10]. The serum level of malondialdehyde (MDA), which is the end product of lipid peroxidation, is a commonly used and recognized biological marker of this process [11].

In our earlier studies, we found some differences in the intensity of lipid peroxidation in patients with CRC depending on selected clinical characteristics; however, the limitation of those studies was the lack of a control group, which is an essential reference point for the evaluation of oxidative stress markers [12,13].

In the present study, serum MDA, TOS, TAS and OSI in patients with CRC and those in the control group were evaluated in order to search for further associations between oxidative stress markers and selected clinical characteristics (i.e., sex, age and body mass index). In addition, an attempt was made at determining the relationship between oxidative stress markers and the location of the primary lesion, clinical stage and histological grade of the disease.

## 2. Materials and Methods

### 2.1. Design, Patients and Samples

The study included 94 patients with CRC (52 men and 42 women) who underwent surgical treatment and none of the patients had oncological treatment (radiotherapy, chemotherapy) before surgery. Control group consisted of 26 patients hospitalized at the Department of Internal Diseases (12 men and 14 women) in whom CRC was excluded based on diagnostic colonoscopy They had no pathological changes in colonoscopy. Patients were enrolled from June 2017 to April 2019.

During hospitalization, blood serum samples (3 cm^3^) were collected from all the patients. Apart from that, anthropometric measurements were taken and body mass index (BMI) was calculated. In addition, the medical records of individuals from the CRC group were analyzed. TNM clinical staging was performed (in accordance with the American Joint Committee on Cancer 8th Edition Cancer Staging Manual, 2017) [14]; the location of the tumour (right side of colon—appendix, cecum, ascending colon, hepatic flexure, 2/3 of proximal transverse colon vs. left side of the colon—1/3 of the distal transverse colon, splenic flexure, descending, sigmoid colon and the rectum) and the histological grade of the disease were also determined.

### 2.2. Inclusion and Exclusion Criteria

Inclusion criteria: 1.CRC adult patients, who underwent surgical treatment min. 4 weeks ago,2.CRC patients who did not receive oncological treatment before surgery,3.CRC patients who gave informed written consent to participate in the study.

Exclusion criteria: 1.patients with previous oncological treatment and/or a current diagnosis of another cancer (except for squamous cell carcinoma of the skin),2.patients with cancer-related cachexia,3.patients requiring permanent immunosuppression,4.patients diagnosed infectious disease, including hepatitis B, hepatitis C, HIV or AIDS.

Due to our intention to determine the relationship between the location of the primary lesion and the studied oxidative stress markers, patients with concomitant lesions in both parts of the colon or lesions of unknown location were excluded from the study. Ultimately, 83 patients were included in the final analysis (47 men and 36 women).

The characteristics of the CRC group (with a breakdown according to sex and location of the primary tumour) and the control group (with a breakdown according to sex) are presented in Table 1 and Table 2. The CRC and control groups were not statistically different in terms of age (*p* = 0.549).

### 2.3. MDA, TOS and TAS Assays

Biochemistry tests were performed on blood collected from fasting patients in the morning (from 8 to 10 h after the last meal). The blood was immediately centrifuged and was stored frozen at −80 °C until the tests were performed. The MDA level was determined using the enzyme-linked immunosorbent assay (ELISA; Cloud-Clone Corp., cat. No CEA597Ge, Houston, TX, USA). TOS and TAS were assayed using the following reagent kits: PerOx (TOS/TOC) (Immunodiagnostik, cat. No KC 5100, Bensheim, Germany) and ImAnOx (TAS/TAC) (Immunodiagnostik, cat. No KC 5200, Bensheim, Germany), respectively, applying the methodology specified by the manufacturer. TOS was determined using a reaction between peroxidase and the total amount of lipid peroxides, while the TAS assay was based on a reaction between the antioxidants in the sample and exogenous hydrogen peroxide (H_2_O_2_). Absorbance readings were recorded at a wavelength of 450 nm, using an ELISA reader (µQuant, BioTek^®^, Winooski, Vermont, USA). Subsequently, calibration was performed according to the standard curve, in ng/mL for MDA and in µmol/L for TOS and TAS. All assays were performed twice.

### 2.4. OSI

In order to determine the level of oxidative stress and redox imbalance, OSI was calculated, which is a ratio between TOS and TAS. The results were presented as a per-centage ratio, based on the following formula: OSI = TOS [µmol/L]/TAS [µmol/L] × 100. Higher ratios in the samples are a sign of predominance of oxidation processes over antioxidant activity [10].

### 2.5. Statistical Analysis

The results were collected in a Microsoft Excel 2010 spreadsheet, while statistical analysis was performed using Statistica 13.3. (TIBCO Statistica™, Kraków, Poland). The normality of quantitative data was determined based on the Shapiro–Wilk test and normality graphs. Quantitative variables with a near-normal distribution were compared using the Student’s *t*-test (two-sample comparison); for quantitative variables with a non-normal distribution, the non-parametric Mann–Whitney U test (two-sample comparison) was used. For the comparison of more than two samples, the Kruskal–Wallis ANOVA on ranks test was conducted. A post hoc Tukey test was also applied. Subsequently, the relationship between the studied oxidative stress markers and selected clinical characteristics was analyzed for both groups. A Spearman’s Rank correlation was calculated between MDA, TOS and TAS between each other and clinical and demographic parameters. A *p* value of < 0.05 was considered statistically significant for all analyses. A receiver operating characteristics (ROC) curve and cutoff value determination using Youden method were performed in easyROC web tool version 1.3. (http://www.biosoft.hacettepe.edu.tr/easyROC/, accessed on 1 April 2020) [15].

### 2.6. Ethics Committee

The study was conducted in accordance with Good Clinical Practice and the Declaration of Helsinki. The study received approval from the Medical University of Silesia, Katowice, Poland, Ethics Committee (protocol code KNW/0022/KB1/43/17 and date of approval 30 May 2017). Participation in the study was voluntary.

## 3. Results

There was no difference between the CRC group and the control group in terms of age (*p* = 0.549). In the majority of the patients, stage III and IV disease was diagnosed. The patients were assessed for adjuvant therapy (chemotherapy [CT] or chemotherapy and radiotherapy [CT-RT]) or palliative systemic treatment. The breakdown of the CRC group according to the clinical stages is presented in Table 3.

The levels of oxidative stress markers: MDA, TOS, TAS and OSI in the CRC group and the control group are presented in Table 4. The evaluation of oxidative stress marker values demonstrated that MDA and TAS levels were significantly higher in the control group than the CRC group (*p* < 0.001 and *p* = 0.019, respectively), while TOS levels were significantly lower in the control group than the CRC group (*p* = 0.005). Significantly lower OSI levels were found in the control group than in the CRC group (*p* < 0.001).

Similar results can be observed when performing ROC analysis of diagnostic ability of oxidative stress markers MDA cutoff value: 4195.49, sensitivity: 0.851 (sensitivity range: 0.763–0.916), specificity: 0.731 (specificity range: 0.522–0.884), AUC = 0.82 ± 0.05, TAS cutoff value: 278.47, sensitivity: 0.702 (sensitivity range: 0.599–0.792), specificity: 0.769 (specificity range: 0.564–0.910), AUC = 0.699 ± 0.07, TOS cutoff value: 428.407, sensitivity: 0.862 (sensitivity range: 0.775–0.924), specificity: 0.538 (specificity range: 0.334–0.734), AUC = 0.72 ± 0.06 and OSI cutoff value: 202.09, sensitivity: 0.777 (sensitivity range: 0.679–0.856), specificity: 0.615 (specificity range: 0.406–0.798), AUC = 0.74 ± 0.06 (see Figure 1). 

Preliminary statistical analysis demonstrated that MDA levels in the CRC group (*n* = 83) depend on the location of the primary tumour (*p* = 0.035). Based on the post hoc Tukey test, a relationship was demonstrated between the MDA level and the left and right side of the colon (*p* = 0.040). No statistical significance was found for TOS, TAS and OSI analysis (see Table 4).

In the CRC group (*n* = 94), a weak positive correlation between MDA level and cancer stage (TNM), rS = 0.21, *p* < 0.05 (see Figure 2).

However, in the control group (*n* = 26), a weak negative correlation between MDA level and age, rS = −0.44, *p* < 0.05 and TAS level and age, rS = −0.597, *p* < 0.05 (see Figure 3).

In the CRC group (*n* = 94), no other differences and correlations were found between the investigated markers of oxidative stress and other analyzed clinical features, such as sex and BMI (*p* > 0.05).

## 4. Discussion

The pathogenesis of CRC has been the subject of ongoing research. Oxidative stress is one of the many factors which can play an important role in carcinogenesis; however, there is no consensus among researchers whether oxidative stress should be considered the cause or the consequence of neoplastic transformation [4]. According to the observations by Leufkens et al. and Thanoon et al., high serum levels of reactive oxygen metabolites can be the result of neoplastic transformation rather than its cause [16,17]. Other researchers maintain that the current status of knowledge does not allow one to find a definitive answer due to the multifactorial nature of CRC and the presence of comorbidities, including metabolic disorders [18].

Numerous in vivo and in vitro studies proved that cancer cells are exposed to continuous and intense oxidative stress compared to cells that are not affected by a neo-plastic process. Redox imbalance can lead to damage to DNA, proteins and lipids [18,19]. Lipid peroxidation is a multi-stage process initiated by the action of free radicals which leads to cellular membrane disruption. Polar peroxide, hydroxyl or aldehyde groups which enter the double internal lipid layer cause its organization to change, leading to its asymmetry. As a result of lipid oxidization, certain membrane enzymes and transport proteins are inhibited. Lipid peroxidation products modify the physical properties of cells’ membranes, which can lead to impaired function of these cells, and, consequently, dysfunction of different organs [7,20]. In numerous studies, a link has been sought between oxidative stress and lipid peroxidation products associated with cancer. Latest evidence suggests that ROS can play an important role at all stages of carcinogenesis, i.e., initiation, promotion and progression [4]. ROS overproduction is responsible for cancer development through, among other processes, genetic mutations, DNA damage, apoptosis inhibition and the promotion of proliferation, differentiation and migration of malignant cells [21]. Many researchers are interested in MDA levels, both in blood samples and cancer tissue. MDA is a compound with a high biological activity, which has cytotoxic, mutagenic and carcinogenic properties [22]. In addition, this tricarbon aldehyde can have an effect not only at its place of origin. It can migrate through cellular membranes, causing harm both in-side and outside of the cells. Unlike free radicals and as the product of their degradation, MDA is characterized by a longer half-life. It can serve as a secondary transmitter of oxidative stress. In addition, MDA is able to modify amino-acid residues, creating stable adducts that destroy proteins. It can also form covalent bonds with DNA and membrane lipids [11,22]. Many studies have confirmed that serum MDA concentration is elevated in patients with malignant neoplasms, including, for example, hepatocellular, renal, urinary bladder, cervical and prostate cancer. Interestingly enough, contradictory results have been observed in CRC patients in this respect [23]. A few studies demonstrated that the MDA level in CRC patients is significantly higher than in control groups [21,22,24,25,26,27]. Furthermore, Zińczuk et al. assumed that MDA may be a potential non-invasive marker used to differentiate between different depths of tumour invasion and an indicator of the presence of metastases in the lymph nodes in patients with CRC [21]. Reports by other authors indicate the presence of lower MDA levels both in blood samples and cancer tissue in CRC patients compared to control groups [28,29]. In the present study, we have also observed such a relationship. Some researchers propose that MDA levels are related to the aggressiveness of the neoplasm in that rapidly dividing cells tend to set the oxidant/antioxidant status to the one conducive to their development [23]. In our study, TOS and TAS were also evaluated. Higher serum TOS levels were observed in patients with CRC than in those from the control group. At the same time, serum TAS levels in the CRC group were found to be lower than those in the control group. Wu et al., (2017) [9], Murlikiewicz et al. (2018) [30] and Zińczuk et al. (2019) [21] obtained similar results in this respect. Lower TAS levels can indicate a compromised antioxidant barrier of the body caused by ROS overproduction. Changed energy metabolism, including the inactivation of certain enzymes, contributes to ROS overproduction in cancer patients. OSI provides information regarding interaction between oxidants and antioxidants. A higher OSI is a sign of predominance of oxidation processes over antioxidant activity [10]. In the present study, OSI was significantly higher in the CRC group than in the control group. The same observations were recorded in papers by other authors [9,21]. Therefore, based on this value, one may conclude that individuals with CRC are at a higher risk of oxidative stress and the associated oxidative damage. In addition, it is believed that the coexistence of oxidative stress and inflammation is an environment that is conducive to the development of CRC. For this reason, patients with metabolic disorders, including obesity, are found to have an increased prevalence of CRC [8]. However, the present study did not demonstrate any relationship between the analyzed oxidative stress markers and BMI in patients with CRC. Nevertheless, it is worth noting the fact that statistically significant differences were observed between serum MDA concentration and the location of the primary lesion in CRC patients. In patients with the tumour located on the right (caecum, ascending colon, hepatic flexure and the proximal 2/3 of the transverse colon), serum MDA levels were higher than in patients with the tumour located on the left (the distal 1/3 of the transverse colon, splenic flexure, descending colon and sigmoid). To date, other authors have not analyzed this aspect; therefore, it seems justified to verify this discovery. In this and in earlier studies, no relationship was observed between TOS and TAS levels (*p* > 0.05) and the location of the primary tumour [13]. The results of the present study need to be corroborated in further research involving a higher number of CRC patients.

## 5. Conclusions

Significant differences were found in MDA levels and primary tumour location. Higher MDA levels were found for the right-sided vs. left-sided colonic location. This suggests the involvement of factors related to lipid metabolism in the development and/or course of proximal colorectal cancer. A weak positive correlation was found between serum levels of MDA and TNM, which may indicate that lipid peroxidation increases with the clinical stage of colorectal cancer. Oxidation dominated over antioxidant effects and lipid peroxidation processes were less intense in patients with colorectal cancer compared to controls. Lower serum MDA levels compared to controls seem controversial and need to be confirmed in further research. In our pilot study, no extended dietary history was taken during the enrolment process, which may be a hindrance to the interpretation of the results. The present authors recommend that the results be treated with caution. Further studies should take into account additional environmental factors affecting the prooxidant-antioxidant balance of the body. Further research might be helpful in the evaluation of diagnostic utility of oxidative stress markers in patients with CRC.

## Figures and Tables

**Figure 1 metabolites-12-01118-f001:**
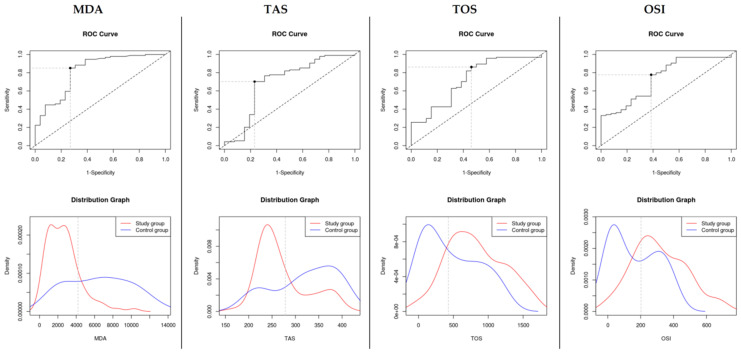
MDA, TAS, TOS and OSI ROC curves and distribution plots with marked cutoff values.

**Figure 2 metabolites-12-01118-f002:**
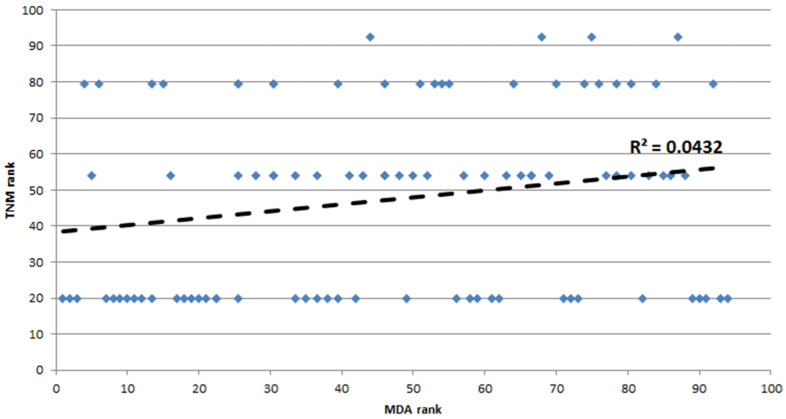
Spearman’s rank correlation between MDA serum concentrations and TNM in the study group.

**Figure 3 metabolites-12-01118-f003:**
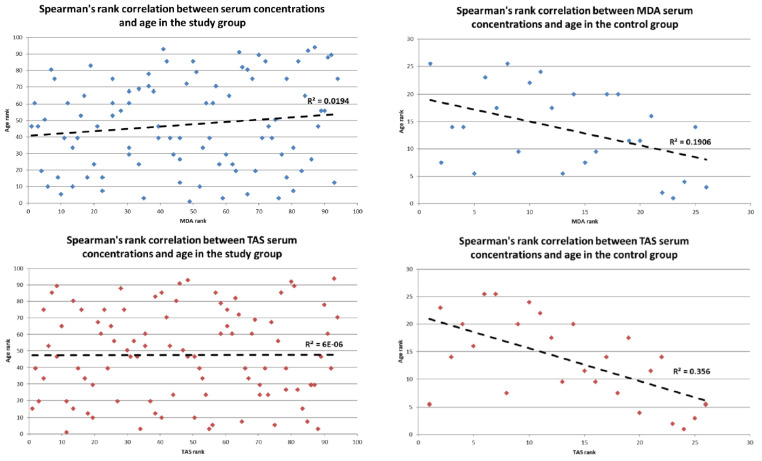
Spearman’s rank correlation between MDA and TAS serum concentrations and age in the study and control group.

**Table 1 metabolites-12-01118-t001:** Characteristics of the CRC group.

Clinical Feature	Mean Age [In Years] ±SD	Mean BMI [kg/m^2^] ±SD	Normal Weight (BMI Value: 18.5–24.9); *n*[%]	Overweight (BMI Value: 25.0–29.9); *n*[%]	Obesity (BMI Value: ≥30.0); *n*[%]
Women (*n =* 42)	66.14 (±11.55)	26.30 (±4.97)	19 [45.24]	12 [28.57]	11 [26.19]
Men (*n =* 52)	63.98 (±9.29)	25.97 (±3.91)	25 [48.08]	18 [34.62]	9 [17.30]
All patients (*n =* 94)	64.95 (±10.36)	26.12 (±4.39)	44 [46.81]	30 [31.91]	20 [21.28]
Left-sided colon (*n =* 32)	64.81 (±11.06)	25.53 (±4.57)	17 [53.12]	12 [37.50]	3 [9.38]
Right-sided colon (*n =* 21)	66.76 (±7.38)	26.70 (±4.38)	8 [38.10]	8 [38.10]	5 [23.80]
Rectum (*n =* 30)	64.57 (±11.46)	26.00 (±4.10)	15 [50.00]	7 [23.33]	8 [26.67]
All colon located tumours (*n =* 83)	65.22 (±10.33)	25.99 (±4.32)	40 [48.19]	27 [32.53]	16 [19.28]

±SD—standard deviation.

**Table 2 metabolites-12-01118-t002:** Characteristics of the control group.

Clinical Feature	Mean Age [In Years] ±SD	Mean BMI [kg/m^2^] ±SD	Normal Weight (BMI Value: 18.5–24.9); *n*[%]	Overweight (BMI Value: 25.0–29.9); *n*[%]
Women (*n =* 14)	67.36 (±9.35)	23.73 (±2.42)	9 [64.29]	5 [35.71]
Men (*n =* 12)	67.83 (±9.89)	24.71 (±1.46)	8 [66.67]	4 [33.33]
Control group (*n =* 26)	67.58 (±9.41)	24.18 (±2.05)	17 [65.38]	9 [34.62]

±SD—standard deviation.

**Table 3 metabolites-12-01118-t003:** Division the group depending on the clinical stages.

Clinical Feature	Clinical Stage (CS)			Grading (G1–G3)			
	CSI, CSII *n*[%]	CSIII *n*[%]	CSIV *n*[%]	G1 *n*[%]	G2 *n*[%]	G3 *n*[%]	Gx *n*[%]
Women (*n =* 42)	13 [30.95]	12 [28.57]	17 [40.48]	5 [11.90]	23 [54.76]	6 [14.29]	8 [19.05]
Men (*n =* 52)	13 [25.00]	17 [32.69]	22 [42.31]	7 [13.46]	29 [55.77]	7 [13.46]	9 [17.31]
All patients (*n =* 94)	26 [27.66]	29 [30.85]	39 [41.49]	12 [12.77]	52 [55.32]	13 [13.83]	17 [18.08]
Left-sided colon (*n =* 32)	7 [21.87]	14 [43.75]	11 [34.38]	4 [12.50]	20 [62.50]	4 [12.50]	4 [12.50]
Right-sided colon (*n =* 21)	6 [28.57]	4 [19.05]	11 [52.38]	4 [19.05]	9 [42.86]	6 [28.57]	2 [9.52]
Rectum (*n =* 30)	10 [33.34]	7 [23.33]	13 [43.33]	4 [13.33]	16 [53.33]	2 [6.67]	8 [26.67]
All patients (*n =* 83)	23 [27.71]	25 [30.12]	35 [42.17]	12 [14.46]	45 [54.22]	12 [14.46]	14 [16.86]

**Table 4 metabolites-12-01118-t004:** Mean (and the lower and upper quartile) MDA, TOS, TAS and OSI serum concentrations in both groups.

Clinical Feature	MDA [ng/mL]	*p*	TOS [µmol/L]	*p*	TAS [µmol/L]	*p*	OSI	*p*
Women (*n =* 42)	2825.38 (1107.77–3679.28)	0.906	865.86 (550.71–1172.70)	0.312	271.23 (231.59–318.27)	0.684	333.13 (218.47–461.86)	0.316
Men (*n =* 52)	2511.15 (1363.10–3332.79)		779.76 (463.87–1114.00)		271.65 (234.06–291.02)		298.92 (176.94–412.92)	
Normal weight (*n =* 44)	2539.27 (849.06–3270.09)	0.222	824.73 (483.44–1224.07)	0.718	279.49 (240.07–309.13)	0.329	313.29 (194.32–456.19)	0.513
Overweight (*n =* 30)	2469.70 (1159.57–3332.79)		762.91 (550.71–925.30)		262.43 (223.31–294.23)		297.35 (228.10–409.07)	
Obesity (*n =* 20)	3171.34 (2052.11–4563.89)		886.93 (487.93–1196.42)		267.35 (234.00–275.47)		341.51 (202.75–467.98)	
CSI, CSII (*n =* 26)	2363.53 (1067.92–3207.38)	0.087	772.09 (454.27–1088.14)	0.794	275.27 (223.31–324.42)	0.741	295.03 (159.27–429.00)	0.970
CSIII (*n =* 29)	2184.48 (1247.52–2922.36)		877.48 (531.84–1185.98)		263.38 (233.33–272.46)		342.40 (218.47–413.69)	
CSIV (*n =* 39)	3190.87 (1645.42–4259.26)		804.94 (464.29–1158.03)		274.93 (232.53–291.55)		306.02 (185.54–450.52)	
All CRC patients (*n =* 94)	2651.55 (1159.57–3332.79)	-	818.23 (510.17–1158.03)	-	271.46 (232.56–293.44)	-	314.20 (203.09–429.00)	-
Left-sided colon (*n =* 32)	2329.20 (1314.10–3270.09)	0.035	812.66 (468.88–1114.00)	0.910	273.49 (232.79–292.36)	0.637	314.08 (202.75–445.43)	0.586
Right-sided colon (*n =* 21)	3565.06 (2186.88–4259.26)		810.12 (590.26–925.30)		269.97 (226.35–318.27)		308.09 (233.25–409.07)	
Rectum (*n =* 30)	2551.27 (1067.92–3269.21)		793.26 (452.77–1185.98)		274.23 (243.35–278.47)		301.33 (165.51–450.52)	
All CRC patients (*n =* 83)	2722.15 (1311.65–3398.18)	-	805.00 (493.40–1130.07)	-	272.87 (232.53–294.23)	-	307.95 (203.09–429.00)	-
Control group (*n =* 26)	6459.85 (2751.18–9743.30)	-	470.63 (111.75–838.83)	-	320.65 (285.90–389.55)	-	164.84(28.14–290.25)	-

## Data Availability

Requests for additional data or for support with reusing the data should be emailed to the authors, who can be contacted at enowakowska-zajdel@sum.edu.pl.

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
