# Peer review of "Evaluation of Malondialdehyde Level, Total Oxidant/Antioxidant Status and Oxidative Stress Index in Colorectal Cancer Patients"

_metabolites, 2022, doi:10.3390/metabo12111118_

Round 1

Reviewer 1 Report

The authors quantified serum malondialdehyde (MDA) level, total oxidant status (TOS), total antioxidant status (TAS) and oxidative stress index (OSI) in the blood samples from 94 CRC patients and 26 controls.

The main findings indicate: 1) an increase of TOS and OSI in CRC patients as compared to controls, despite MDA was found at higher levels in controls; 2) an increased MDA concentration (on 83/94 patients) in patients with right side CRC, as compared to left side tumors.

As a general comment, the first finding was already reported in several papers (as stated by the authors in the discussion), while the second finding was already reported by the same group a couple of years ago. Thus, the present study lacks of novelty, also being unable to give a final contribution to the controversies reported in the discussion.

I found some problems also in the methods and some tables/figures are incomplete, or difficult to interpret. I kindly ask the authors to provide an upgraded version of the paper to be able to correctly interpret its contents. I also suggest additional analyses, to improve the results.

Methods

The information about blood sampling is insufficient. It appears a random sampling of a completely heterogeneous cohort: other authors of similar studies sampled only pre and/or post (within 48h) surgery samples (see ref 9 of the paper as an example). According to the inclusion/exclusion criteria of this manuscript, patients bearing the primary CRC, or attending adjuvant therapy (post-surgery? = no more tumor?), or palliative therapy (inoperable/diffused tumor?) were recruited, without a shared timepoint for sampling (i.e. beginning/end of therapy cycles). As therapy can indeed induce systemic stress and a pro-oxidant reaction, I would expect a strong background signal, unrelated to CRC itself.

Data on the levels of MDA and other markers, related to the therapeutic regimen at the time of sampling, should be added in table 4 (or in an additional table if preferred).

Please report in tables 1,2,4 not only the mean value, but also the range (or standard deviation) of showed parameters.

Please, report as a supplementary file the pathologies of control patients (they are not healthy donors), as it could be relevant for radical production and explain the strange distribution of MDA and TAS values according to age.

Results

I’ve great problems in interpreting the Y axis of figure 2. Figure 2 would report the TNM rank, but TNM would create 9 subcategories (corresponding to stages II-A, II-B, II-C, III-A, III-B, III-C, IV-A, IV-B, IV-C), instead I see four categories. If TNM has been used as a synonym of a simplified stage grouping I-IV, yet I would expect to see three parallel lines of MDA values as no stage I patient was recruited. Please, edit the graph according to the extended TNM staging of tumors (it will also help statistics), possibly creating an Y axis reporting nominal variables. As a better alternative, I would suggest the switch of axes and a box and whiskers representation, with Kruskal-Wallis test.

Right sided CRC is frequently late-diagnosed due to the absence of symptoms. This allows the tumor to reach higher volumes and form necrotic/ulcerated areas.  Both increased tumor dimension and presence of necrosis (causing neutrophils infiltration and activation), could participate to the observed plasmatic increase of MDA. The authors should add a new analysis correlating MDA levels to the tumor volume and the presence of ulceration/necrosis, and/or leucocytes infiltration (these data are usually available in standard pathology reports). This analysis could be very important, perhaps explaining why right-side CRC shows higher MDA levels.

If the authors are willing to add new experimental data, I would also suggest to test residual blood samples (if available) for one inflammatory marker (i.e. TNFα, or IL1, or IL-8), as it could link the pro-oxidant burst to immune activation.

Another explanation for the increased MDA levels in right side CRC, could be the presence of prevalent aerobe bacteria in this district. These bacteria are the main sources of short chain fatty acids and produce MDA.  The authors could analyze if serum MDA levels are increased when occluding tumors of the upper right district are present (near the hepatic flexure in particular). In this case, the reduced mobility of fecal content could destabilize bacterial scavenger metabolism and increase MDA accumulation.

The authors of ref 9, reported that the levels of TOS, TAS, and OSI were significantly different between patients with no metastasis and those with metastases to two organs with 2 or more localizations, do you have data confirming this observation?

Thank you.

Reviewer 2 Report

Manuscript iD: metabolites-1961720

Authors: Janion et al.,

In this research article entitled “Evaluation of malondialdehyde level, total oxidant/antioxidant status and oxidative stress index in colorectal cancer patients”, the authors studied the relationship between oxidative injury, clinical stage/grading and the location of the primary lesion in colorectal cancer (CRC) patients.

Hereafter, some points that should be taken into account before processing further.

Comments to the authors:

-          In the abstract, the sentence “The evaluation of oxidative stress… the study group (P=0.005)”, in lines 21-24, should be rewritten as

i)                    it is complicated and,

ii)                   in fact, the reported parameters:  MDA, TAS and TOS are all higher in the CRC group.

-          The authors used the expression “the study group” a couple of times, starting by the abstract itself in line 23. I guess, it would be better to replace it by “the CRC group”.

-          What does “ROC”, in line 26, stand for? The full name should be given once the abbreviation is firstly used.

-          What do the authors mean by “left and right sides of the colon”? Please specify using anatomical terminology: ascending, transverse, descending… parts of the colon? Please clarify to the readers.

-           The authors should mention in sentence (line 55), that free radicals are the consequence of both physiological AND ALSO pathophysiological processes.

-          It would be better to change “including chronic ones”, in line 64, by “including CRC” as the study focus on CRC patients.

-          Table 1 and 2 may be merged. In fact, both exhibit the characteristics of the studied groups (CRC and control).

-          A consort flow diagram may be helpful for better and easier understand of the inclusion/exclusion criteria, patients and the study design.

-          Regarding English language, it is fine but minor checking is required for both spelling and punctuation. Such as “(ROS), which” instead of “(ROS) which” in line 55, “stages” instead of “stage” in line 167, and “were” instead of “was” in line 167, “in” instead of “ between” in line 294…

Round 2

Reviewer 1 Report

I thank the authors for their attempt to complete their study.

While my doubts for the possible bias in sampling have been resolved, most open questions remain without answer and the results do not introduce any novelty. Moreover, reading for the second time the revised manuscript, I noticed in methods that blood sampling was almost parallel to the first study published in 2020 (June 2017 - December 2018, vs June 2017 - April 2019). This would suggest that the present study contains the old cohort, added with 28 new cases and some controls. This approach would void the statistical validation of the first 2020 data with a second cohort, as it is not independent. Moreover, knowing the involvement of right colon, the new cohort should have been enriched with right side CRC and could avoid rectal CRC samples. The focus on a specific stage could also improve the paper (I would suggest stage II CRC, to limit the systemic modulation linked to lymph node or liver metastases).

The controls are not sufficiently defined: even if any colorectal lesion has been excluded by colonoscopy, an internal medicine department collects patients with several morbidities that can affect the redox balance; moreover, no age matching has been reported, while CRC is typically a cancer of elderly people.

I realize, from the answer letter, that the authors are not able to obtain further clinic information on their cohorts, thus it is impossible for them to improve the manuscript. Unfortunately, I cannot accept the study in the present form. I’m really sorry for this decision.

Best regards

Reviewer 2 Report

Manuscript metabolites-1961720 (Round #2)

After providing a revised version along with a response to reviewer’s comments, we can obviously notice that the manuscript has been improved and may be accepted for publication in metabolites.
